# Federated Learning-Based Spectrum Occupancy Detection

**DOI:** 10.3390/s23146436

**Published:** 2023-07-16

**Authors:** Łukasz Kułacz, Adrian Kliks

**Affiliations:** 1Institute of Radiocommunications, Poznan University of Technology, 60-965 Poznan, Poland; lukasz.kulacz@put.poznan.pl; 2Department of Computer Science, Electrical and Space Engineering, Luleå University of Technology, 971 87 Lulea, Sweden

**Keywords:** federated learning, machine learning, spectrum occupancy detection

## Abstract

Dynamic access to the spectrum is essential for radiocommunication and its limited spectrum resources. The key element of dynamic spectrum access systems is most often effective spectrum occupancy detection. In many cases, machine learning algorithms improve this detection’s effectiveness. Given the recent trend of using federated learning, we present a federated learning algorithm for distributed spectrum occupancy detection. This idea improves overall spectrum-detection effectiveness, simultaneously keeping a low amount of data that needs to be exchanged between sensors. The proposed solution achieves a higher accuracy score than separate and autonomous models used without federated learning. Additionally, the proposed solution shows some sort of resistance to faulty sensors encountered in the system. The results of the work presented in the article are based on actual signal samples collected in the laboratory. The proposed algorithm is effective (in terms of spectrum occupancy detection and amount of exchanged data), especially in the context of a set of sensors in which there are faulty sensors.

## 1. Introduction

Recently, the popularity of radiocommunication system utilization has been growing. Many new and more demanding services are expected to be delivered wirelessly with guaranteed and typically high quality. Each smartphone, laptop and even smartwatch, as well as many IoT devices, cars, etc., require access to the wireless network, which leads to the very challenging problem of limited spectrum resources. For many reasons, the statically and exclusively allocated spectrum is only partially utilized. It is common that the spectrum resources are not used all the time or even are not used at all, as certain services are not delivered in the given location. In view of the problem of insufficient spectrum resources, dynamic spectrum access (DSA) is a very promising solution to meet the growing requirements for radiocommunication systems [1,2,3]. Contrary to the static-allocation approach, DSA can use the unoccupied spectrum for purposes other than those for which it was initially intended and reserved. Although conceptually simple, practical realization of the DSA is not a trivial task. By using the appropriate spectrum-sensing algorithms, unlicensed users (also known as secondary users, SUs) should have access to new spectral resources, which are detected as unused at the given time and location [1,4]. DSA may be realized in the wide-range form, but also in the small-scale variant [5]. At the same time, care should be taken to ensure that the primary user’s (PU) transmission is provided with the best possible quality of service, whether through appropriate separation of resources between users, control of the transmission power of system users or even disabling the possibility of transmission for unlicensed users. The key aspect here is to gain or collect information about the presence of primary users’ transmissions. Once the system has this information, it can take appropriate action to protect the PU’s transmission. It is necessary to have a well-established understanding of the spectrum hole [6]. The straightforward and most popular way of collecting information about spectrum occupancy is spectrum sensing [7,8]. Many solutions have been proposed in the rich literature in this respect, exploring various opportunities in the process of determining PU signal presence. One may mention such approaches as energy detection, feature detection and cyclostationary-based spectrum sensing, to name a few. However, single-node spectrum sensing cannot guarantee the required level of reliability of the spectrum, thus cooperative variants are widely considered, where collaborating nodes exchange information (raw measurements, rough decisions, final decisions, etc.) to improve sensing performance. In such an approach, both distributed and centralized (i.e., with the presence of a fusion center) approaches are possible [9,10]. On a big scale, the term crowd sensing is used [11], where multiple sensing nodes collaborate. However, one of the key problems related to cooperative spectrum sensing (CSS) is the huge amount of data to be exchanged between the cooperating nodes. It is also necessary to identify the spatial correlation between the cooperating nodes as it should be clear if the sensing information from a specific node is also valid for another node.

The paper is organized as follows. In the next subsection, we describe related work and then in Section 2 we revise the foundations of the FL and describe the overall FL-based spectrum-sensing scheme. In Section 3, we present the experimentation setup, discuss the features detected during the FL spectrum-sensing process and present the reference scenario. In Section 4 and Section 5, we analyze the results achieved for the spectrum occupancy detection in one and over multiple channels. All results presented in the paper are outlined in the conclusion and discussed in Section 6.

### Related Work

Furthermore, machine learning (ML) algorithms are also successfully used to improve the quality of detection, which supports the detection process by extracting important but not always noticeable details of the transmission [12]. An example of such details may be the periodicity of (or within) the transmission, different traffic volumes depending on the time of day or year or the characteristics of the transmission system used. One of the solutions for spectrum-sensing efficiency improvement is to benefit from various predictions in time, space and frequency domains, which can be characterized by, e.g., daily traffic demand trends. In such an approach, some ML techniques could be used to predict such trends, like recurrent neural networks (RNN) [13] or support vector regression (SVR) [14]. Various traffic patterns can be detected in different domains and they appear very often simultaneously using advanced AI/ML solutions [15,16,17]. However, the application of ML towards better spectrum sensing has some limitations; for example, the usage of supervised learning entails the need for labeling reference signals and this may be problematic. In addition, as we mentioned above, the knowledge from a single sensor can often be distorted due to the sensor’s specific location or its unreliability. Recently, distributed learning (in its various variants) has gained attention in wireless communications domains. Thus, cooperative spectrum-sensing algorithms have been considered an improvement, yet they lead to increased complexity in ML algorithms [18].

One of the promising solutions to the above problems is applying the federated learning (FL) concept in the spectrum-sensing procedure. Here, regardless of whether the FL variant is used (i.e., centralized or distributed), the nodes share the artificial intelligence (AI) model instead of raw data to detect the presence of PU better. Such an approach has many advantages—the overall control traffic is reduced while benefiting from the cooperation between the nodes [19,20]. In particular, there are multiple works related to FL in radiocommunication systems; they, however, consider aspects other than spectrum occupancy detection. Such FL algorithm could be utilized, e.g., in resource allocation process [21]. The summary of FL algorithm application for wireless networks can be found in [22]. In terms of the clue of this paper, there are not that many publications that utilize FL for spectrum sensing. In [23] authors were focused on waveform detection in the citizens broadband radio service (CBRS) system and they showed that FL can improve detection efficiency. Then in [24], the authors presented the FL framework which could be used in cooperative spectrum sensing (CSS). In our work, we focus more on the FL application in the real environment. In the as-found signal received by the sensors, we propose an algorithm for spectrum occupancy detection, which achieves comparable accuracy to classical approaches and is resistant to the presence of faulty sensors. In [25], a very interesting, exhaustive and complete work describing long-term research on the FL framework can be found. The authors performed the whole FL process with many adjustments, however, the authors worked on an air quality dataset, which cannot be directly compared to the spectrum-sensing challenge. A crucial part of FL and ML algorithms is a clustering algorithm, especially in the context of distributed spectrum sensing. In another interesting work [26] FL algorithm is proposed mainly for the classification of road objects in the vehicular to everything (V2X) network.

Motivated by the above observation, we wanted to address the important problem of reliable distributed spectrum sensing realized by resource-limited devices (sensors). The idea was to propose a solution that, to some extent, benefits from the experience of the neighboring nodes and minimizes the amount of shared data between them. The advantage of FL in that context lies in the fact that sensing nodes may in some sense share the gained knowledge with other nodes about the environment in the form of the model coefficients.

In this paper we propose the utilization of centralized FL for spectrum sensing in the scenario with three sensors. The proposed solution assumes machine learning model coefficients exchange between these sensors—in a central server and the creation of a common model that is further used by each sensor. A couple of machine learning models were examined and a comparison with the reference (without FL) scenario was done. Additionally, a case with a faulty sensor was verified. The proposed solution also focuses on the main bottleneck of spectrum sensing in commercial networks—rare access to verified information—which could be used as training data. This paper presents the results of the hardware experiment where FL has been implemented in the laboratory, where universal software radio peripheral (USRP) boards have acted as sensing nodes.

## 2. Federated Learning Spectrum Sensing (FLSS)

Reliable spectrum sensing, especially in a harsh environment (like operation in very low signal-to-interference-plus-noise, SINR, regime), usually requires a very long sensing time or the cooperation of numerous nodes [27,28,29]. Consequently, there is an immediate problem of either a long sensing time or a vast amount of data that must be transferred between the sensors. In both cases, the time available for the actual transmission of user data is reduced and so is the system’s overall efficiency. It should be remembered that sensors should also be very simple devices, which is an additional limitation regarding the number of data transferred between them. Another critical challenge is the transparency of the transmitted data, i.e., the transmission of the collected measurement data by sensors can be relatively easily intercepted and analyzed and, worse, replaced.

One of the promising ideas for solving these problems that is gaining popularity recently and is widely considered in many aspects and applications is federated learning (FL) [30]. By assumption, instead of sharing raw data, the local nodes share between themselves only the parameters specifying the ML models (such as coefficients of the neural networks). In the classical approach, the central node trains the global model and shares it with the underlying nodes, which receive the global model and adjust it locally based on its local observations. Once the ML models residing in the local nodes are trained (and thus adjusted to the local scenario), they are sent back to the central node to update the global model and spread the updates within the whole cooperating network. The whole process is then repeated, allowing continuous learning and adaptation of the ML models (both global and local). The above procedure may be summarized as follows. First, one has to specify the set of cooperating nodes and define the learning parameters such as ML model details, learning rate, number of communication rounds, the local batch size for updating local models or an algorithm used for combining old and new models. Let us assume that there are *K* cooperating nodes indexed by k=1,…,K and each node *k* updates the weights of the local ML model, denoted as $x_{k}$. Then, the ultimate optimization function will be denoted as $f\left(x_{1},\ldots ,x_{K}\right)$ and may be calculated as the average of local objective functions $f_{i}\left(x_{i}\right)$:(1)$$f\left(x_{1},\ldots ,x_{K}\right)=\frac{1}{K}\sum_{i=1}^{K}f_{i}\left(x_{i}\right).$$

This simple averaging approach may be modified to reflect the typical situation where local nodes may operate over different sets of samples (of cardinalities denoted as $n_{k}$).

Inspired by [19,25], in our work, we concentrate on applying the FL in a cooperative sensing network, where a set of nodes exchange model coefficients to sense the wireless spectrum better and more accurately. Hereafter, we refer to FL spectrum sensing (FLSS). As stated before, FLSS assumes the operation of individual spectrum sensors in a specific area of the network. They collect data by observing the spectrum in their location and use these data to train their ML model. Then, the SS model coefficients of individual sensors are exchanged, as opposed to the samples collected in traditional cooperative spectrum sensing and an aggregated model is created. To this end, various merging approaches may be considered, such as the simplest Federated Averaging (FedAvg) [31] scenario—created by averaging all coefficients received from the sensors—or Federated Stochastic Gradient Descent (FedSDG), which is based on gradient computation over a random subset of the whole dataset and uses them to make one step of the gradient descent. Once created, the coefficients of the collective model are transferred to all sensors. In the simplest scenario, the sensors use a collective model and may update it based on the newly collected spectrum samples. The procedure described above is repeated many times. As a consequence, individual sensors have access to a common SS model, which is at least indirectly based on many more samples than those collected by a single sensor. One of the benefits of using FLSS is the ability to train a single sensor model much faster in the event of its replacement or simply adding a new one. Therefore, the speed of adaptation of such a system should be its advantage—especially in the context of the aforementioned general challenge related to spectrum occupancy detection (rare access to verified information—training data). Similarly, the ability to detect a faulty sensor (i.e., by detecting unexpected reports from it) should be a valuable safeguard for the system leading to higher sensing reliability. Obviously, FLSS used in this way cannot offer higher detection efficiency than cooperative spectrum occupancy detection algorithms due to limited access to training data (only indirectly). However, at the expense of slightly lower detection efficiency, it is possible to significantly reduce the amount of sent control data and improve system security.

In this work, we aim to verify the efficiency of hardware-implemented centralized FLSS, where the sensing nodes observe the environment in their vicinity, update the local model and share the model coefficients with the central node. The central node applies the FedAvg algorithm and distributes the models among the cooperative nodes. Once the local nodes receive the new global model, they average it with its local copy and continue sensing with the updated one. As the ML technique used in the FL procedure, mainly the neural network has been selected. The considered solution is presented graphically in Figure 1. Additionally, we compare the results with the non-FL approach, where nodes apply locally some of the ML techniques (mainly support vector machine, K-nearest neighbors and random forest), but do not share the models’ coefficients with the surrounding nodes. In the following, we present the setup of the conducted experiment.

## 3. Experimentation Setup

Let us now discuss the experimentation setup used in the laboratory to verify the practical performance of the FL-based spectrum sensing. First, we present the overall hardware setup and present the process of data collection. Next, we discuss the implemented ML tool for FLSS, as well as the whole process of data preparation for reliable reasoning. After that, we present the reference scenario.

### 3.1. Measurement Setup

As we concentrated on the evaluation of the FLSS performance in the real environment, it was necessary to gather a rich set of real samples and work on them. To achieve this, the true measurement data were collected.

As illustrated in Figure 2, three measurement series were made, each of which was carried out for a different position of the signal receiver (spectrum sensor). Each of the analyzed receiver positions varied between 2 and 5 m from the transmitter, at a similar height (difference less than 15 cm) and with direct visibility between the transmitter and receiver antennas. During all three measurements, the position of the signal transmitter remained constant. The signal transmitter and receiver were implemented with the help of a laptop with Ubuntu and GNU Radio software 3.8 and USRP connected to the computer via USB 3.0 interface. The signal generated by the transmitter was in the form of six adjacent Gaussian Minimum Shift Keying (GMSK) signals with a bandwidth of 350 kHz each. Each of the signals was placed in the frequency domain in a separate channel with a width of 1 MHz. The center frequency of the transmitter is 2100 MHz; hence, the next carriers of the GMSK signal are 2097.5, 2098.5, 2099.5, 2100.5, 2101.5 and 2102.5 MHz, respectively. The spectrum of the transmitted GMSK comb signal is shown in Figure 3.

For each sensor location, in-phase and quadrature samples of the received signal were collected—first with the transmitter turned off (3 times 10M samples) and then with the transmitter turned on (31 times 1M samples), each time changing the gain setting of the transmitter amplifier. The transmit amplifier gain settings used were from 35 dB to 80 dB. The receiver was set to a center frequency of 2100 MHz, a bandwidth of 40 MHz. The receiver was prepared to record samples from 40 adjacent channels, 1 MHz each. However, taking into account the non-linear nature of the filters and amplifiers in the physical receiver, additional processing had to be performed. First, the local oscillator frequency shift was set to 10 MHz and second, the extreme channels (12 MHz on each side) were omitted in the analysis process. The local oscillator frequency shift was intended to minimize the constant power observed at the carrier frequency of the receiver. The extreme channels were omitted due to the imperfections of the physical receiver.

### 3.2. Data Analysis and Preparation

Once the raw data are collected (i.e., the samples representing the observed environment), it has to be decided how they will be processed for reliable decision-making. In the case of FLSS, the ultimate goal is to detect the presence or absence of the PU signal in the observed spectrum. As in classic spectrum sensing, the simplest solution is calculating the observed signal power. However, the experiments have shown that relying only on the received signal power in the ML algorithm would not be sufficient to obtain better results than the classical energy-detection scheme. To this end, the collected data were subjected to the following analysis. For each analyzed channel, a portion of samples equal to 10k was processed, based on which three values were computed: average power, kurtosis of the autocorrelation function and skewness of the autocorrelation function. In this way, it was noticed that the data obtained for these two cases (i.e., presence and absence of PU transmission) are clearly distinguishable for all three features; the signal strength is obviously higher when there is an active PU transmission, but kurtosis and skewness are lower. This observation confirms the correctness of selecting these three properties of the observed signal for use in ML algorithms. Obviously, the conclusion drawn above is valid and the distinction between the presence and absence of PU signal is easy when one operates in the high signal-to-noise regime (i.e., the transmitted signal is much stronger than the noise). However, for a low signal-to-noise ratio case, this task is not trivial and it is expected that also, in that case, the proposed FLSS should work efficiently. Let us also note that in data processing, the collected data for all three sensors were also normalized depending on the requirements of the ML algorithm; scaling to a specific range was the most common approach. All data used for ML were published in this public repository [32].

### 3.3. Reference Scenario

To evaluate the performance of the proposed solution, we compared it with traditional spectrum sensing and with non-FL approaches. In the former case, the energy-detection algorithm, which consists of several simple steps, was adopted as the reference scenario. In the latter case, we compare the FLSS with the non-FL logistic regression model and neural networks (with two hidden layers containing four neurons each). In the energy-detection case, the estimation of the noise power $\sigma^{2}$ is required for the collected data (for each sensor separately). Second, for the given false alarm probability $p_{fa}$ (in this paper, $p_{fa}=0.01$), the detection threshold Γ was calculated according to the following formula [8]:(2)$$\Gamma =\sigma *\left(Q\left(p_{fa}\right)\sqrt{\frac{2}{N}}+1\right),$$
where *Q* is the tail distribution function of the standard normal distribution and *N* is the number of samples (*N* = 10,000 in this paper). The last step is to decide about the presence Ψ of the PU signal by comparing the received power with the calculated detection threshold:(3)$$\Psi =\left\{\begin{array}{cc} 1 & \mathrm{if}\;P_{rx}\;\geq \;\Gamma \\ 0 & \mathrm{otherwise} \end{array}\right.$$
where $P_{rx}$ is power of the received signal. The correctness of the collected results was determined by the model’s accuracy. Mainly, it was verified by the number of data entries that were predicted (by the model) correctly in terms of its label. As a label in this case, we used information about the presence or lack of the transmitted signal. Data were prepared using the *StratifiedKFold* method and thus divided into five parts. This results in an even division in each of the parts between the data with and without the presence of the transmitted signal. In this case, the training data were used for noise estimation and the verification data for the actual signal detection. For the three sensors used in the experiment, we obtained an average accuracy of 0.7846, 0.8621 and 0.6564; the average accuracy among all sensors was 0.7677. A difference between accuracy score achieved for particular sensors is mainly due to the different distances between the transmitter and receiver (i.e., the source of the signal and the sensor). It results in different SNRs and in more ”samples” with higher received power by some sensors (which is easier to classify as a transmitted signal—not noise).

Similarly, non-FL solutions were analyzed in terms of their accuracy. The accuracy scores achieved by all sensors are presented in detail in Figure 4. The verification of the logistic regression model and the neural network (carried out in exactly the same way as for classic energy detection) showed average accuracy of 0.8, 0.9968 and 0.7029 (with an average between sensors equal to 0.8332) for logistic regression and 0.799, 0.9991 and 0.7178 (with mean between sensors equal to 0.8386) for the neural network. This shows that using additional information, such as kurtosis and skewness of the autocorrelation function, allows for an average accuracy improvement over the energy-detection scheme. In particular, it improved by 6.55 percentage points when using the logistic regression algorithm and 7.09 percentage points when using the neural network.

Once the accuracy of the reference scenarios has been checked, let us now evaluate the performance of the proposed FLSS scheme. Thus, in the next section, we will focus on the analysis of a single channel (of the six analyzed) to facilitate the presentation of concepts and algorithms. The following sections will describe how the cumulative algorithm works with all six channels.

## 4. Discussion—Single Channel Spectrum Sensing

The federated learning algorithm proposed in this paper assumes the independent operation of all three sensors. I.e., although they share model coefficients between themselves, they collect their own data and learn their own machine learning model, and make independent decisions solely. Immediately before deciding to detect the spectrum, each sensor sends its model coefficient to the central node. In the central node, a common model is created, which in the following step is distributed among all sensors. Next, the sensors make a decision based on a common machine-learning model. The joint model is created in the central node as a weighted average of the coefficients of each model obtained from the sensors. The weight for each sensor is based on its last accuracy. Values less than 0.4 are ignored. This is to reject the model’s coefficients, which in the previous trial gave poor results. The experiment was carried out in the following way. First, we treat all data from all sensors separately. The datasets were shuffled; next, the first 100 datasets were sent to the sensors for initialization of the local ML models and for noise estimation required in the reference model. The simulator then runs iteratively; during each iteration, the sensors processed 30 sets of data of *N* = 10,000 samples. Each sensor fitted the model to the received data and sent the trained model to the central node, creating a common model; once computed, the global model was distributed. Once the training phase was finished, the testing phase was initiated. Thus, each sensor receives a set of test data (the remaining data of a specific sensor not used during the learning process) and verifies the accuracy of its own model.

As a result, each sensor obtains slightly worse results than if the model coefficients were not exchanged. This is very logical behavior because the data from neighboring sensors are different and a sensor tailored to its own data would achieve better results than when it also has to consider data from other sensors. However, in the other part of the verification, the models were checked by all the data that were used in the simulation (regardless of whether they come from measurements from a given sensor or another) as shown in Figure 5. In this case, much better results were obtained than with single sensor data. This indicates that a created common model is better suited to the general data emerging in this environment. It allows, for example, for easier replacement of sensors and faster adaptation in the event of failure of one of them. This is particularly important in the context of the main problem of spectrum occupancy detection, i.e., access to reliable information about the presence of a signal (later used as a data label in the machine learning process).

It is worth taking a closer look at the results obtained when one of the sensors deliberately reports wrong decisions in the process of FLSS. In this experiment, we assumed these would be arbitrary decisions about spectrum occupancy. Indeed, the data used to train individual models are slightly different for different sensors. This may be due to the specific location of the sensor or even the use of another device. When exchanging model coefficients between sensors, it is necessary to compensate for these differences in the inputs of the machine learning algorithms. Otherwise, the averaging (or other mathematical operation) of the coefficients of models with different input data characteristics will diminish the models’ effectiveness.

## 5. Discussion—Multi-Channel Spectrum Sensing

In the previous section, we concentrated on an evaluation of the results of the single-channel FLSS; we would like now to extend it to the multi-channel case. Thus, in the next simulation stage, all steps (from data preparation through model training, validation and federated learning simulations as described) were repeated for six adjacent channels of the GMSK-comb signal presented in Figure 3. Each channel was treated independently, i.e., each channel has its own FLSS model and its own data and in the process of federated learning, the coefficients of independent models were exchanged separately for each channel. In this way, six separate common models were created. A fundamental assumption in the considered system is the specificity of the transmission, which assumes the use of all six channels at once or none of them. In the last phase, when the decision on the spectrum occupancy is made, the decision criterion of at least four out of six were used, i.e., at least four channels must be considered occupied to consider that the entire analyzed band (six adjacent channels) is occupied. Referring to classical cooperative spectrum sensing, such an approach is known as majority voting [33].

The result of model validation for the reference scenario (presented in Figure 6, Figure 7 and Figure 8) was the average accuracy score (average between trials and between analyzed channels) for all involved sensors: 0.7891, 0.8595 and 0.7364 (mean between sensors 0.795). Three indicated figures show the mean accuracy score of each sensor in the scenario, where the detected signal is transmitted in all six channels at once. Bars described as ”TH” represent the energy-detection algorithm; those denoted ”SVM”, ”kNN” and ”RF” show the accuracy score of support vector machines (SVM), K-nearest neighbors (kNN) and random forest (RF), respectively. The last two sets of bars present results for logistic regression (LR) and dense neural network (DNN). The reference scenario using logistic regressions in the machine learning process obtained average accuracy for the three involved sensors at the level of 0.8086, 0.9968 and 0.7715 (with an average between sensors equal to 0.859). In the reference scenario using a neural network in the machine learning process, the following accuracy was achieved: 0.7912, 0.9661 and 0.7647 (with an average between sensors equal to 0.8406).

For the federated learning algorithm, the following average accuracy was obtained (average between trials) assuming the selected decision criterion (i.e., minimum four out of six occupied channels) and logistic regression for common model 0.7730. The average accuracy for the neural network is 0.8945. In the reference scenario where the decision is made only based on energy detection, the value is 0.8708. This set of results is presented in Figure 9, Figure 10 and Figure 11. These figures show the mean accuracy score of each sensor in the scenario, where the detected signal is transmitted in all six channels at once. Models used in this simulation are the same as in previously described figures (and are denoted in the same way, as “TH”, “SVM”, “kNN”, “RF”, “LR” and “DNN”). Here, a group of six bars for each model does not present the accuracy score in each individual channel, but alternatively we take into account the knowledge that the signal was transmitted in each channel simultaneously. For this reason, we added majority voting (*k* of *n*, in our case we considered all variants, i.e., from 1 to 6 ending with from 6 to 6) and the accuracy score achieved for each voting method is presented in the form of bars.

Assuming that one sensor is generating random decisions during the simulation, we will observe two important effects. First, the obtained average accuracy equals 0.7268. This was achieved while averaging results over all experimentation trials, assuming the selected majority decision-making criterion and logistic regression used for the common model. Analogously, the average accuracy for the neural network case was at the level of 0.8395. Second, for the same simulation scenario, where one sensor is faulty but model coefficients are not exchanged, we observe the average accuracy for logistic regression of 0.2787, 0.7504 and 0.7704 per each sensor with an average between sensors of 0.5998. The neural network’s average accuracy was equal to 0.3095, 0.6998 and 0.8773 (average between sensors 0.6289). As a result, if logistic regression is utilized with federated learning, the average accuracy score can be improved by 12.7 percentage points in cases when one faulty sensor is observed. If a neural network is utilized, then the average accuracy score can be improved by around 21 percentage points.

## 6. Conclusions

The algorithms presented in the work show that there is a potential for using federated learning in spectrum occupancy-detection systems. In specific scenarios, they provide increased reliability of detection and, more importantly, ensure greater efficiency of decisions made compared to classic algorithms (such as energy detection) and comparable efficiency compared to algorithms using machine learning (compared to a model trained on a larger number of samples—such as in cooperative spectrum occupancy detection). It would be worth considering checking other model and result validation metrics, such as the F1 metric. Using different ML techniques could be beneficial in different scenarios. In the considered scenario, if logistic regression is utilized with federated learning, the average accuracy score was slightly improved in cases when one faulty sensor is observed. The amount of data (coefficients) that needs to be exchanged remains at a low level, as this ML model uses only a few coefficients. When a neural network is utilized, then the average accuracy score can be improved more than in the logistic regression case, however with the cost of around ten times more coefficients needed in the model coefficient exchange procedure. This leads us to the point where we can choose the most suitable ML model for the considered scenario and network. In future work, we plan to make a comparison between presented results and the distributed system, where the model’s coefficients are exchanged only with the central node but between neighboring sensors. Such a distributed approach should be more efficient in terms of the data amount that needs to be exchanged and should have better local decisions—especially in the network places over huge areas. However, we suppose the algorithm of model creation will be the most critical. Due to this, the presence of faulty sensors will be more challenging.

## Figures and Tables

**Figure 1 sensors-23-06436-f001:**
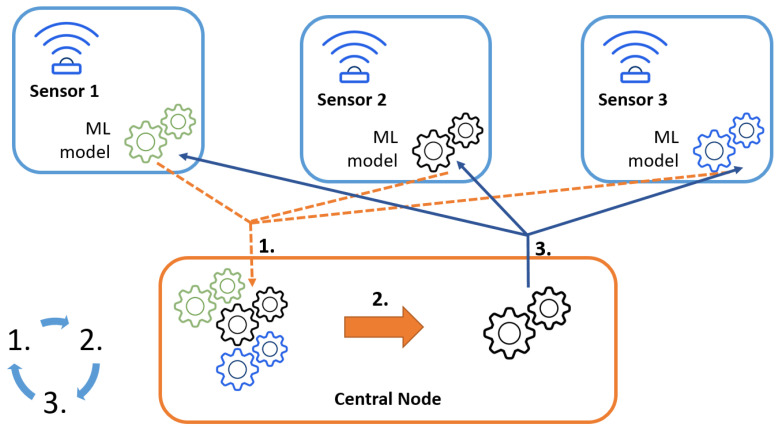
Considered system solution.

**Figure 2 sensors-23-06436-f002:**
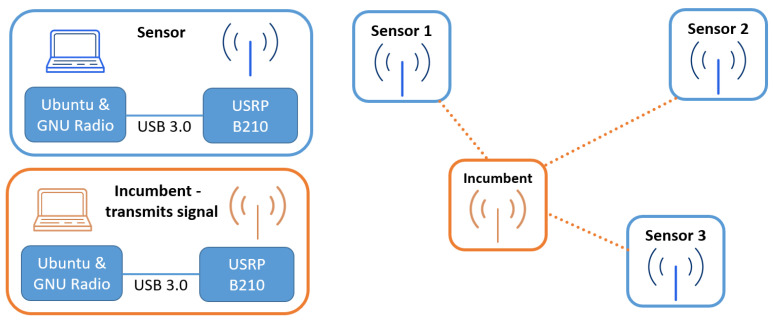
Measurement setup.

**Figure 3 sensors-23-06436-f003:**
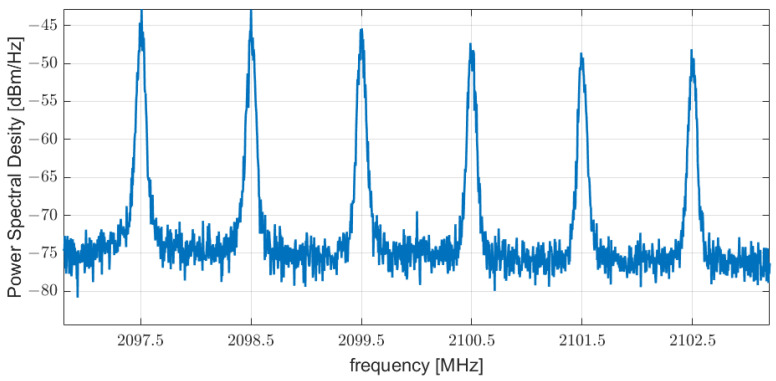
The spectrum of transmitted GMSK comb signal.

**Figure 4 sensors-23-06436-f004:**
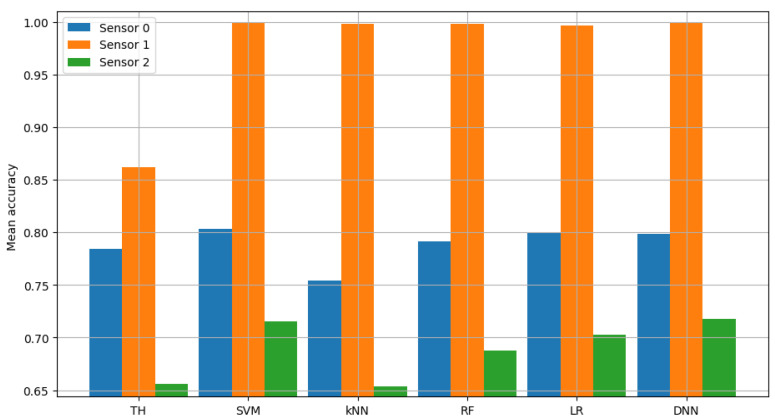
Validation of single channel model.

**Figure 5 sensors-23-06436-f005:**
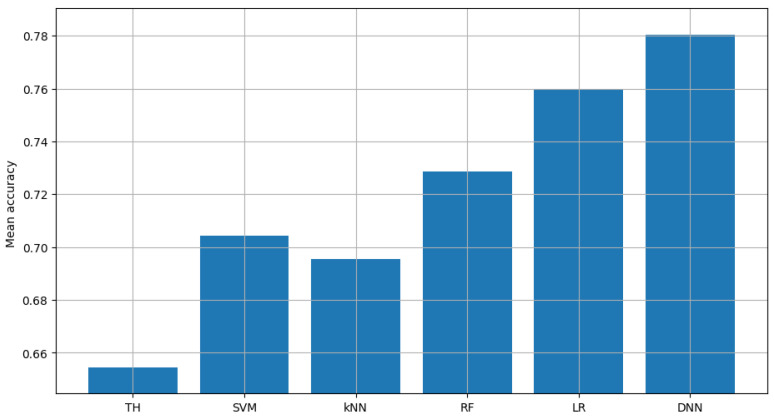
Accuracy score of single channel model (federated learning).

**Figure 6 sensors-23-06436-f006:**
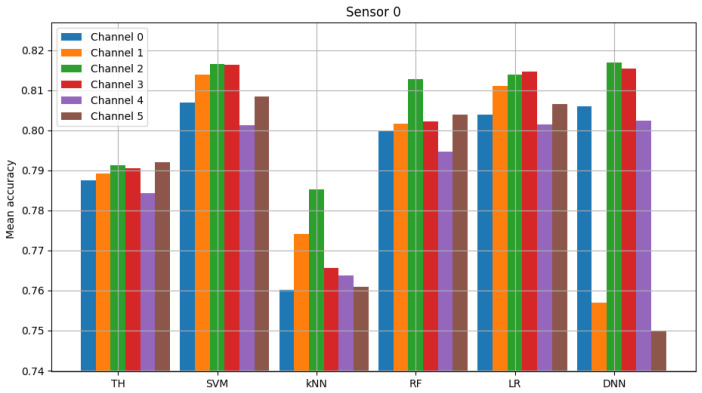
Validation of “sensor 0” model (multiple channels).

**Figure 7 sensors-23-06436-f007:**
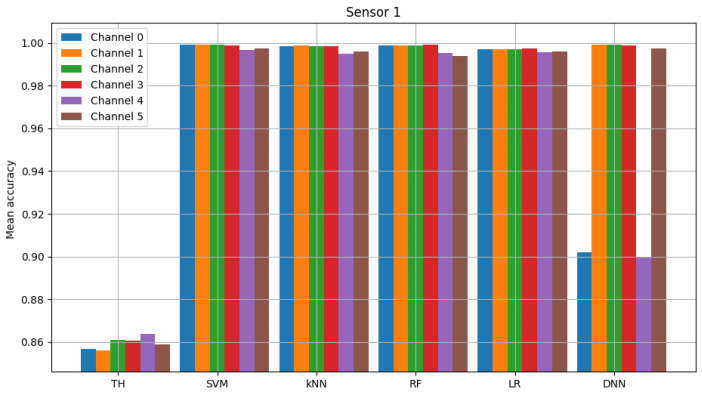
Validation of “sensor 1” model (multiple channels).

**Figure 8 sensors-23-06436-f008:**
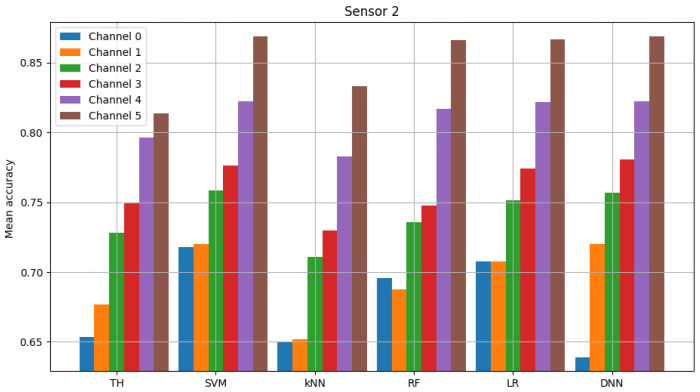
Validation of “sensor 2” model (multiple channels).

**Figure 9 sensors-23-06436-f009:**
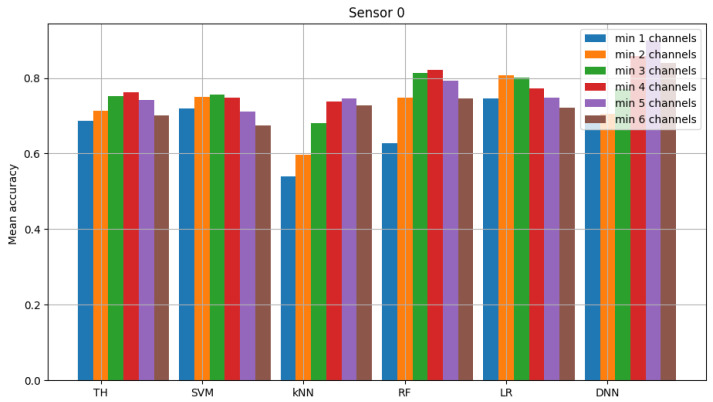
Accuracy score of sensor 0 model (multiple channels and federated learning).

**Figure 10 sensors-23-06436-f010:**
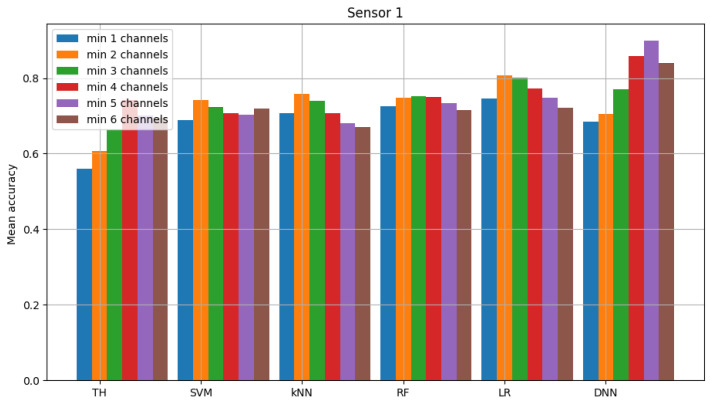
Accuracy score of sensor 1 model (multiple channels and federated learning).

**Figure 11 sensors-23-06436-f011:**
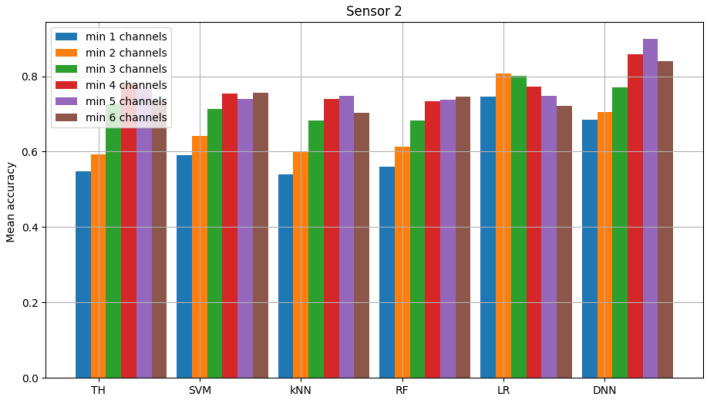
Accuracy score of sensor 2 model (multiple channels and federated learning).

## Data Availability

The data presented in this study are openly available in Zenodo at 10.5281/zenodo.8046418.

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
