# Peer review of "Federated Learning-Based Spectrum Occupancy Detection"

_sensors, 2023, doi:10.3390/s23146436_

Round 1

Reviewer 1 Report

The work is good, but some sections of the work are missing.

1. In the introduction section, the contributions of this work should be emphasized (it should also be specified what this paper proposes).

2 A section with related work is missing and a comparison with other solutions from the specialized literature is missing. 

3. The conclusion section is missing.

Author Response

Responses are inside PDF file.

Reviewer 2 Report

In this study, the authors propose a federated learning algorithm for distributed spectrum occupancy detection. The results of the work presented in the paper are based on actual signal samples collected in the laboratory. The idea is interesting. Some comments are as follows: 

1. The abstract is too short. One feels that the highlights of this article are not highlighted. The abstract needs to be written a bit more to highlight its own contribution.

2. What is the core contribution of the paper?

3. What are the bottlenecks in this area?This brings the research motivation of this paper.

4. In terms of literature research, it is suggested to add the description for the following machine learning work: A fuzzy cluster validity index induced by triple center relation. IEEE Transactions on Cybernetics

5. What is the theoretical contribution of this paper? For example, in the core theory section, there should be more formulas and explanations to explain the core idea of the method proposed in this paper.

6. Why are the issues of a federated learning algorithm for distributed spectrum occupancy detection. studied in this article important?

7. Abstract: The proposed algorithm is effective, especially in the context of a set of sensors in which there are faulty sensors    -->  The proposed algorithm is effective, especially in the context of a set of sensors in which there are faulty sensors.

8. Some explanation needs to be added to Figures 6-11.

9. The future work outlook related to Federated Learning needs clarification. This helps make the context of the study clearer to the reader.

10.There is a significant lack of literature in the last 3 years, and there is also a lack of literature in the top journals in the field.

The English expression of the paper needs to be optimized.

Author Response

Responses inside PDF file

Reviewer 3 Report

Comments to the Author

This paper presents a federated learning algorithm for distributed spectrum occupancy detection. The paper touches on an interesting topic. However, there are several points that need to be addressed to improve the quality of the manuscript.

Suggestions to improve the quality of the paper are provided below:

1)     The Introduction section starts very abruptly starting with “In view of the problem of insufficient resource…”, without providing any general context about radiocommunication systems, or why there is a problem of insufficient spectrum resource. Consider starting the Introduction section with some contextual information.

2)     Many of the background information behind federated learning and FLSS that was covered in the first two paragraphs of Section 2 should be shifted up to the Introduction section when it was first introduced to improve the paper’s readability.

3)     Please clearly state the objective and list out the contributions of this work to clearly show how this work extends upon the existing literature in the Introduction section. The current objective seem to only state what is done in this paper.

4)     Section 2 currently contains some background information about federated learning and FLSS, some literature review on FLSS, and the paper’s methodology of the proposed system. I suggest the authors to shift the background information about federated learning and FLSS to the Introduction section when it was first introduced, rename Section 2 as literature review, and create Methodology section.

5)     The literature review section is currently very lacking with no reference of past studies that adopted a FLSS approach. It would also be good to include a short review talking about similar applications where FL was adopted.

6)     Figure 4 shows the accuracy of different sensors using different ML algorithms based on a single channel model. However, there seems to be a very distinct performance difference between different sensors. Why is that the case?

7)     It is uncommon to end the manuscript with a Discussion section. Instead, please divide the current Discussion section into two separate sections: Discussion and Conclusion.

8)     In the new Discussion section, please discuss about the limitations of this work and how it will be addressed in future works. For instance, given the simple experiment setup adopted in this study (i.e., laboratory setting, three sensors placed close to each other and with direct visibility), how does the authors intend to expand on this study to improve the generalisability of the study’s findings?  

9)     Minor comments on paper structure and writing

·       Consider replacing the results in Figures 6-11 with a table to allow easy comparison between different sensors, machine learning models, and learning modes

The paper was written in a clear manner with minor spelling mistakes. Please go through the manuscript thoroughly and correct them.

Author Response

Responses inside PDF file

Round 2

Reviewer 1 Report

The work was substantially improved. The authors addressed all the reported issues.

Author Response

Thank you very much for all valuable comments.

Reviewer 3 Report

Thank you for taking the time to address my comments. I have no additional comments and believe the manuscript is ready for publication.

Author Response

(The authors gave the same response as above.)
